# Genome-Wide Survey for Microdeletions or -Duplications in 155 Patients with Lower Urinary Tract Obstructions (LUTO)

**DOI:** 10.3390/genes12091449

**Published:** 2021-09-20

**Authors:** Luca M. Schierbaum, Sophia Schneider, Stefan Herms, Sugirthan Sivalingam, Julia Fabian, Heiko Reutter, Stefanie Weber, Waltraut M. Merz, Marcin Tkaczyk, Monika Miklaszewska, Przemyslaw Sikora, Agnieszka Szmigielska, Grazyna Krzemien, Katarzyna Zachwieja, Maria Szczepanska, Katarzyna Taranta-Janusz, Pawel Kroll, Marcin Polok, Marcin Zaniew, Alina C. Hilger

**Affiliations:** 1Institute of Human Genetics, University Hospital of Bonn, 53127 Bonn, Germany; luca.schierbaum@uni-bonn.de (L.M.S.); sophia.schneider@uni-bonn.de (S.S.); sherms@uni-bonn.de (S.H.); julia.fabian@uni-bonn.de (J.F.); reutter@uni-bonn.de (H.R.); 2Human Genomics Research Group, Department of Biomedicine, University of Basel, 4031 Basel, Switzerland; 3Institute for Medical Biometry, Informatics and Epidemiology, Medical Faculty, University of Bonn, 53127 Bonn, Germany; sugi@uni-bonn.de; 4Institute for Genomic Statistics and Bioinformatics, Medical Faculty, University of Bonn, 53127 Bonn, Germany; 5Core Unit for Bioinformatics Data Analysis, Medical Faculty, University of Bonn, 53127 Bonn, Germany; 6Department of Neonatology and Pediatric Intensive Care, University Hospital Erlangen, 91054 Erlangen, Germany; 7Department of Pediatrics, University Hospital Marburg, 35033 Marburg, Germany; stefanie.weber@med.uni-marburg.de; 8Department of Obstetrics and Prenatal Medicine, University of Bonn, 53127 Bonn, Germany; waltraut.merz@ukbonn.de; 9Department of Pediatrics, Immunology and Nephrology, Polish Mother’s Memorial Hospital Research Institute of Lodz, 93-428 Łódź, Poland; marcin.tkaczyk45@gmail.com; 10Department of Pediatrics, Cardiology and Immunology, Medical University of Łódź, 93-428 Łódź, Poland; 11Department of Pediatric Nephrology and Hypertension, Jagiellonian University Medical College, 31-007 Krakow, Poland; mmiklasz@wp.pl (M.M.); katarzyna.zachwieja@gmail.com (K.Z.); 12Department of Pediatric Nephrology Medical University of Lublin, 20-059 Lublin, Poland; sikoraprzem@hotmail.com; 13Department of Pediatrics and Nephrology, Medical University of Warsaw, 02-091 Warsaw, Poland; aszmig@yahoo.com (A.S.); grazyna.krzemien@litewska.edu.pl (G.K.); 14Department of Pediatrics, School of Medicine with the Division of Dentistry in Zabrze, Medical University of Silesia in Katowice, 40-055 Katowice, Poland; szczep57@poczta.onet.pl; 15Department of Pediatrics and Nephrology, Medical University of Białystok, 15-089 Białystok, Poland; katarzyna.taranta@wp.pl; 16Neurourology Unit, Pediatric Surgery and Urology Clinic, 61-701 Poznań, Poland; pawelkroll.poczta@gmail.com; 17Neurourology Unit, Poznan University of Medical Sciences, 61-701 Poznań, Poland; 18Department of Pediatric Surgery and Urology, University of Zielona Góra, 65-417 Zielona Góra, Poland; polok.m@gmail.com; 19Department of Pediatrics, University of Zielona Góra, 65-417 Zielona Góra, Poland; m.zaniew@wlnz.uz.zgora.pl

**Keywords:** lower urinary tract obstruction (LUTO), posterior urethral valves, male limited phenotype, de novo, copy number variations (CNVs), maternal transmission

## Abstract

Lower urinary tract obstruction (LUTO) is, in most cases, caused by anatomical blockage of the bladder outlet. The most common form are posterior urethral valves (PUVs), a male-limited phenotype. Here, we surveyed the genome of 155 LUTO patients to identify disease-causing CNVs. Raw intensity data were collected for CNVs detected in LUTO patients and 4.392 healthy controls using CNVPartition, QuantiSNP and PennCNV. Overlapping CNVs between patients and controls were discarded. Additional filtering implicated CNV frequency in the database of genomic variants, gene content and final visual inspection detecting 37 ultra-rare CNVs. After, prioritization qPCR analysis confirmed 3 microduplications, all detected in PUV patients. One microduplication (5q23.2) occurred de novo in the two remaining microduplications found on chromosome 1p36.21 and 10q23.31. Parental DNA was not available for segregation analysis. All three duplications comprised 11 coding genes: four human specific lncRNA and one microRNA. Three coding genes (*FBLIM1*, *SLC16A12*, *SNCAIP*) and the microRNA MIR107 have previously been shown to be expressed in the developing urinary tract of mouse embryos. We propose that duplications, rare or de novo, contribute to PUV formation, a male-limited phenotype.

## 1. Introduction

Congenital lower urinary tract obstructions (LUTO) are a heterogeneous group of pathologies caused by anatomical blockage of the bladder outflow tract or by functional impairment of urinary voiding. The most common form of LUTO are posterior urethral valves (PUV), a male-limited phenotype [1,2]. This phenotype presents in 63% of patients with LUTO [2]. Furthermore, urethral stenoses and urethral atresia are found, which can occur in both sexes [3] (Figure 1). 

LUTO have an estimated birth prevalence of three in 10,000 [4]. Severe cases present prenatally with megacystis, while milder forms can first manifest during childhood [3]. The intrauterine urethral obstruction can lead to damage of both kidneys, and even terminal kidney failure, due to prenatal urinary retention. The reduced urinary excretion into the amniotic cavity results in oligo- or anhydramnios with consecutive pulmonary hypoplasia and joint contractures, and can potentially lead to all of the features of the Potter sequence [3]. These secondary problems can be prevented in some patients by vesico-amniotic shunting, thus relieving the urinary retention and ensuring a sufficient amount of amniotic fluid. However, there are also patients who develop end-stage renal failure postnatally despite successful vesico-amniotic shunting [5], suggesting that either structural renal abnormalities or very early kidney damage occurring before shunting could be causative. Thus, according to the literature, 20% to 65% of PUV patients will develop chronic kidney disease (CKD) during childhood with 8–21% developing end-stage renal disease (ESRD), which makes it one of the most common causes of ESRD during childhood [6]. If untreated, 45% of the patients die in the perinatal period [2]. For anatomical blockages, so far only variants in *BNC2* have been described as causative [7]. Previous studies outlined the role of causative genetic copy number variations (CNVs) in a substantial proportion of patients with congenital anomalies of the kidneys and the urinary tract (CAKUT) [8]. While for LUTO patients, possible potential causative CNVs could only be described in individual patients up until now [9], previous studies identified an enrichment for rare duplications in PUV patients compared to controls [10]. To identify potential disease-causing genes, we systematically surveyed the genome of 155 LUTO patients. 

## 2. Materials and Methods

### 2.1. Subjects and DNA Isolation

This study was approved by the local ethics committee (Lfd. Nr. 031/19). Written informed consent was obtained from each family prior to their inclusion in the study. The 155 LUTO patients comprised 122 patients with PUV and 23 LUTO patients with an anatomical blockage of uncertain cause (unclear if it is PUV or stenosis). All families were recruited in Germany and Poland. All families were recruited within the CaRE for LUTO Study (Cause and Risk Evaluation for LUTO) by experienced physicians in Germany and Poland. Primary inclusion criteria are either a known prenatal condition of megacystis or the postnatal diagnosis of LUTO. All LUTO patients underwent a thorough clinical examination. In the case of reported urinary tract infections or of voiding anomalies in first-degree relatives, an ultrasound study followed, and if necessary, an uroflowmetry was performed in order to rule out or identify LUTO. DNA of all LUTO patients and 4.392 healthy in-house controls were extracted from blood or saliva samples using the Chemagic Magnetic Separation Module I (Chemagen, Bäsweiler, Germany) or, in the case of saliva samples, the Oragene DNA Kit (DNA Genotek Inc., Kanata, ON, Canada). 

### 2.2. Array Genomic Hybridization Analysis

For molecular karyotyping, we used the Illumina Infinium Global Screening Array-24v1.0 plus Multi-Disease add-on content Bead Chip (marker content, 642,824; median marker spacing 4.59 kb). A DNA sample was considered to have failed quality control if (i) less than 98% of the markers were generated on the respective BeadChip, (ii) the called sex did not correspond to the biological sex, and (iii) the total number of CNVs per sample ≥ the double standard deviation of CNVs per patient.

### 2.3. CNV Analysis

#### 2.3.1. Quality Check

To identify potential CNVs, the SNP fluorescence intensity data from all patients with LUTO and all controls were analyzed with three different calling programs: (i) QuantiSNP (v2.1 and v2.2, www.well.ox.ac.uk/QuantiSNP/ (accessed on 14 May 2018)) [11], (ii) PennCNV [12], and (iii) cnvPartition (Illumina, Inc., San Diego, CA, USA). Whereas cnvPartition runs in GenomeStudio and implements a partitioning approach, QuantiSNP and PennCNV apply the Hidden-Markov-Model on the exported LogR ratio (normalized intensity data) and B allele frequency data (allele frequency data). All three algorithms were applied to the data of one GenomeStudio project, which holds genome-wide data of all cases and controls used in this study, to reduce processing and normalization bias.

On the identified CNVs, the following quality criteria were applied to exclude CNVs: (i) log Bayes factor < 30, and (ii) regions with ≤ 3 aberrant probes. 

#### 2.3.2. Filter Algorithms 

**Physical overlap (i)** In order to exclude CNVs that were present in our in-house control sample and as part of a frequency control, we developed a filter algorithm that encountered physical overlap of each CNV within the CNVs called in our in-house control sample. All CNVs overlapping exactly or completely within >1 of the in-house controls were excluded. Duplications and deletions were filtered independently and separately for each calling program. Only CNVs called in all three programs were included in further studies. Information on gene or promotor content in the remaining aberrations was added by AnnotSV [13]. This tool annotates and ranks structural variants including CNVs based on various databases such as the 1000 Genomes Project and dbVar. In addition, AnnotSV generates gene-based annotations using RefSeq gene symbols and reports whether a gene is partially or fully affected by a CNV. AnnotSV also provides a classification system according to the ACMG guidelines and reports on the pathogenicity of each CNV call. Only CNVs spanning a gene- or promotor-coding region have been included in further studies. **Deleteriousness (ii)** According to the Database of Genomic Variants (DGV), < 10% of all CNVs reported are larger than 50 kb, leading to the hypothesis that larger CNVs potentially cause more damage [14]. Therefore, we used a size filter excluding all CNVs < 50 kb. CNVS were checked for their presence in the Database of Genomic Variants (DGV) (http://dgv.tcag.ca/dgv/app/home (accessed on 16 September 2018)). Only CNVs without any completely or exactly overlapping CNVs reported were included for further study. **Visual inspection (iii)** All filtered CNVs underwent visual inspection using the GenomeStudio genotyping module (V2.0.5, www.illumina.com/ (accessed on 15 October 2018)) for marker and signal quality (Appendix A). **Prioritization (iv)** CNVs were prioritized based on literature research (https://pubmed.ncbi.nlm.nih.gov (accessed on 15 April 2019)). We gathered information with a focus on functional and expressional studies as well as reported urorectal phenotypes associated with potential candidate genes residing in the CNV regions. Embryologic mouse expression data (http://www.informatics.jax.org (accessed on 16 April 2019)) were used to prioritize genes with high expression in cloacal/urethral/bladder tissue in mouse embryos. All data are designated to hg19, and the identified de novo CNV has been deposited in ClinVar (https://www.ncbi.nlm.nih.gov/clinvar/ (accessed on 7 September 2021)).

### 2.4. CNV Validation Using Quantitative PCR (qPCR)

For validation of the filtered CNVs, qPCR was performed, as described previously, using SYBR Green for detection [15].

## 3. Results

In total, we validated three microduplications in three independent LUTO patients (Table 1). Among these, we identified a de novo 852,734 bp microduplication at 5q23.2 in a male PUV patient. On chromosomes 1p36.21 (65,758 bp) and 10q23.31 (131,276 bp), we identified one microduplication each, which was not reported in DGV or our in-house control sample. All three patients presented with PUV. Parental DNA samples for segregation analysis were not available, leaving it uncertain whether these microduplications occurred de novo or not.

The confirmed de novo duplication on chromosome 5q23.2 comprised four human specific non-coding RNAs (LINC02201, LOC101927357, LOC105379152, and MGC32805/PPIC) and four coding genes (*PRDM6*, *SNCAIP*, *SNX2*, *SNX24*). *Sncaip* has been found to be expressed at mouse embryonic day 15.5. (E15.5) in the developing kidney [15]. The other three coding genes show no expression in the mouse embryonic urinary tract system. Furthermore, almost no data exists for the embryonic role of the four long-coding RNAs.

Microduplication 1p36.21 comprised *FBLIM1*, *PLEKHM2*, *SLC25A34*, *SLC25A34-AS1*, and *TMEM82*. Interestingly, mouse *Fblim1* is strongly expressed at E15.5 in the embryonic mouse urethra and other embryonic urinary tract structures [16]. All other coding genes comprised by this microduplication show no expression in the mouse embryonic urinary tract system. 

Microduplication 10q23.31 comprises the human microRNA MIR107 and the two human coding genes *PANK1* and *SLC16A12*. *SLC16A12* has been shown to be expressed in the early embryonic mouse kidney at E17.5 [16]. Interestingly, *SLC16A12* has also been discussed as a potential biomarker for human prostate cancer [17]. While there has not been any association reported between LUTO and prostate cancer to date, PUVs typically localize in the verumontanum, a structure located on the floor of the posterior urethra, which marks the boundary between the membranous and the prostatic segment [18]. Independently, Mir107 has been shown to be expressed at E12.5 in the embryonic mouse metanephros [16]. Moreover, human MIR107 has been reported to exert pleiotropic functions in human bladder and prostate cancer [19].

## 4. Discussion

Here, after analyzing the genome of 155 LUTO patients, we identified three novel microduplications. One microduplication, 5q23.2, occurred de novo, and for the two remaining microduplications found on chromosomes 1p36.21 and 10q23.31, parental DNA was not available for segregation analysis. 

Among the identified microduplications, the microduplication 5q23.2 that occurred de novo has the highest likelihood of being disease-causing. The human coding gene *SNCAIP* is located within this microduplication. This de novo duplication on chromosome 5q23.2 comprises four human specific non-coding RNAs (LINC02201, LOC101927357, LOC105379152, and MGC32805/PPIC) and four coding genes (*PRDM6*, *SNCAIP*, *SNX2*, *SNX24*). Among the four coding genes, *Sncaip* has been found to be expressed in early embryonic mouse kidneys [16]. In a large genome-wide association study on diabetic kidney disease, Salem et al. [20] were able to show that variant rs149641852 at chromosomal position 5:121774582 within *SNCAIP* is associated with extreme chronic kidney disease (defined as eGFR < 15 mL/min/1.73 m^2^), dialysis or renal transplant for the “CKD extreme” phenotype. The DECIPHER [21] database has listed an entry for a confirmed novel de novo variant on chromosomal position 5:122451506-122451506 (GRCh38, C > G). The patient’s phenotype is described to show abnormalities of the head, the eyes, the musculoskeletal system, the nervous system, and the genitourinary system. More interestingly, Giardino et al. [22] described a male child with a recombinant chromosome 5, resolved by FISH analyses to have a 5q23.2–31.3 duplication, inherited from his phenotypically normal mother who carried a balanced ‘pericentric’ insertion of these bands in 5p13.1. Remarkably, the male index patient was born by Cesarean section after 38 weeks of gestation, after the pregnancy had been complicated by oligohydramnios. At four months of age, the child exhibited hypotonia, microcephaly, hypertelorism, cranial asymmetry, a prominent occiput, low-set ears, retrognathia, a hypoplastic mandible, a club foot and partial syndactyly of the second and third toes of both feet. His cardiac defects included ventricular hypertrophy and hyperkinesias. Pulmonary hypoplasia and stenosis, bilateral hydronephrosis, hydrocele, testicular hypoplasia and phimosis were also observed. The oligohydramnios, the facial appearance resembling that of a child with Potter sequence, and the pulmonary hypoplasia accompanied by bilateral hydronephrosis highly suggest that their index patient had some form of anatomical LUTO, although this was not described by the authors. Hence, our current findings, together with the previous reports on *SNCAIP*, propose that duplications of chromosomal band 5q23.2 comprising *SNCAIP* and single variants within *SNCAIP* are involved in genitourinary tract anomalies or chronic kidney disease. 

To the best of our knowledge, we could not detect previous reports on microduplications 1p36.21. Interestingly, distal 1p36 duplications have been associated with rectal stenosis and/or anterior displacement of the anus (https://www.orpha.net (accessed on 15 July 2019)). From an embryonic perspective, the final bladder outlet and anorectum both originate from the primitive cloaca during the sixth and seventh week of embryonic development (Carnegie stages 15–23) [23]. This common origin and the previous reports on the occurrence of rectal stenosis and/or anterior displacement of the anus associated with distal 1p36 duplication warrant further exploration of this genomic region and its possible involvement in urorectal anomalies.

To the best of our knowledge, we could not detect previous reports on microduplications 10q23.31. Nevertheless, as outlined above, *SLC16A12* has been shown to be expressed in the early embryonic mouse kidney at E17.5 [16] and has been discussed as a potential biomarker for human prostate cancer [17]. Moreover, microdeletions of chromosomal band 10q23.31 encompassing *SLC16A12* and *PANK1* have directly been associated with prostate cancer [24]. Since most PUVs localize directly in the prostatic urethra, the association of SLC16A12 with prostate cancer seems interesting. Interestingly, another member of the Solute Carrier Family, *SLC20A1,* has been suggested to be involved in cloacal malformations and kidney cysts in humans, supported by zebrafish studies [25]. Independently, Mir107 has been shown to be expressed at E12.5 in the embryonic mouse metanephros [16] and was found to exert pleiotropic functions in the human bladder and prostate cancer [24].

## 5. Conclusions

Overall, several lines of evidence suggest that the novel microduplications identified here are involved in the formation of LUTO, particularly PUV. This observation is in line with the previous reports of an enrichment of genomic duplications in LUTO patients.

We propose that systematic molecular karyotyping in larger LUTO cohorts will identify further causative CNVs of which a substantial proportion will be de novo in origin. Due to the fact that this is, in most cases, a male-limited phenotype, maternal transmission from healthy mothers must be kept in mind.

## Figures and Tables

**Figure 1 genes-12-01449-f001:**
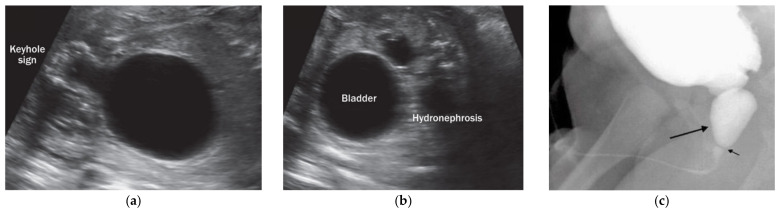
The phenotype of LUTO. (**a**,**b**) Prenatal fetal ultrasound with typical “keyhole sign” of the obstructed bladder with congenital LUTO; (**c**) postnatal VCUG (voiding cystourethrography) of a patient with congenital PUV (long arrow pointing at pre-stenotic urethra, short arrow pointing at urethral valve).

**Table 1 genes-12-01449-t001:** The table presents the three validated microduplications with length, mode of inheritance and affected genes.

Patient ID	Phenotype	CNV	Length	Mode of Inheritance	Affected Genes
100243	PUV	dup5q23.2	852.7 kbp	de novo	LINC02201/LOC101927357/LOC105379152 /MGC32805/PPIC*/PRDM6/* *SNCAIP/SNX2/SNX24*
100295	PUV	dup10q23.31	131.3 kbp	no parental DNA	MIR107*/PANK1/SLC16A12*
100009	LUTO	dup1p36.21	65.7 kbp	no parental DNA	*FBLIM1/PLEKHM2/SLC25A34/* *SLC25A34-AS1/TMEM82*

CNV, Copy Number Variation; PUV, Posterior Urethral Valve; LUTO, Lower Urinary Tract Obstruction; dup, duplication; kbp, kilo base pairs.

## Data Availability

All data are available upon request. The identified de novo CNV has been submitted to clinVar (SUB10219848).

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
