# Peer review of "Genome-Wide Survey for Microdeletions or -Duplications in 155 Patients with Lower Urinary Tract Obstructions (LUTO)"

_genes, 2021, doi:10.3390/genes12091449_

Round 1
Reviewer 1 Report
The consequences of Lower Urinary Tract Obstructions, especially the problem related to the presence of the posterior urethral valves in boys, are extremely dangerous and often lead to chronic renal failure. Every research attempting to explain the mechanisms of these diseases is very valuable, especially if it entails the extension of diagnostic possibilities and the rapid initiation of treatment.
Author Response
We thank the reviewer for his kind commentary on our study.
Reviewer 2 Report
This is a very stringent paper describing microdeletions or duplications in patients with Lower Urinary Tract Obstructions (LUTO), and should be accepted after minor modifications are conducted.
In the Introduction section, more data should be given on certain forms of LUTO. For example, epidemiological studies show that posterior urethral valves are seen in 63% of all LUTO (please consult Cheung et al. Best Pract Res Clin Obstet Gynaecol 2019;58:78-92. doi: 10.1016/j.bpobgyn.2019.01.003).
In the Materials and Methods section of the paper, more information on how families were recruited should be provided. Literature research for prioritization (in the "2.3.2. Filter algorithms" part of the manuscript) should also be explained in more detail.
Discussion may be broadened to increase the scope of the manuscript. For example, papers like Rieke et al. SLC20A1 Is Involved in Urinary Tract and Urorectal Development. Front Cell Dev Biol. 2020;8:567 may be included.
Conclusion and Outlook should be merged into one section.
It is not clear why a historic reference from 1969 is used as a reference 1 for a claim that posterior urethral valves are the most common form of LUTO. Abeit not a mistake per se, a newer reference would be more fitting in my opinion.
Author Response
Comment 1:
This is a very stringent paper describing microdeletions or duplications in patients with Lower Urinary Tract Obstructions (LUTO), and should be accepted after minor modifications are conducted. In the Introduction section, more data should be given on certain forms of LUTO. For example, epidemiological studies show that posterior urethral valves are seen in 63% of all LUTO (please consult Cheung et al. Best Pract Res Clin Obstet Gynaecol 2019;58:78-92. doi: 10.1016/j.bpobgyn.2019.01.003).
Answer to Comment 1:
We thank the review for the comments on our study. As requested we edited the introduction section and added the sentences “This phenotype presents in 63% of patients with LUTO [2].” and “If untreated 45% of the patients die in perinatal period [2]“.
Comment 2:
In the Materials and Methods section of the paper, more information on how families were recruited should be provided. Literature research for prioritization (in the "2.3.2. Filter algorithms" part of the manuscript) should also be explained in more detail.
Answer to Comment 2:
The Materials and Method section on family recruitment now reads: “All families were recruited within the CaRE for LUTO Stuady (Cause and Risk Evaluation for LUTO) by experienced physicians in Germany and Poland. Primary inclusion criteria are either a known prenatal condition of megacystis or the postnatal diagnosis of LUTO. All LUTO patients underwent a thorough clinical examination. In the case of reported urinary tract infections or of voiding anomalies in first degree relatives, an ultrasound study followed and if necessary, an uroflowmetry was performed in order to rule out or identify LUTO.”
We explained the prioritization process based on the literature research in more detail. Section 2.3.2. Prioritization iiii) now reads as:
“Prioritization iv) CNVs were prioritized based literature research (https://pubmed.ncbi.nlm.nih.gov) gathering information with a focus on functional and expressional studies as well as reported urorectal phenotypes associated with potential candidate genes residing in the CNV regions. Embryologic mouse expression data (http://www.informatics.jax.org) were used to prioritize genes with high expression in cloacal/urethral/bladder tissue in mouse embryo. All data are designated to hg19, the identified de novo CNV has been deposited in ClinVar (clinvar@ncbi.nlm.nih.gov).”
Comment 3:
Discussion may be broadened to increase the scope of the manuscript. For example, papers like Rieke et al. SLC20A1 Is Involved in Urinary Tract and Urorectal Development. Front Cell Dev Biol. 2020;8:567 may be included.
Answer to Comment 3:
We added a sentence on the Solute Carier Family: “Interestingly, another member of the Solute Carrier Family, SLC20A1, has been implicated to be involved in cloacal malformations and kidney cysts in human, supported by zebrafish studies [25].”
Comment 4:
Conclusion and Outlook should be merged into one section.
Answer to Comment 4:
The merged Conclusion and Outlook section now reads as:
“Overall, several lines of evidence suggest that the here identified novel microduplications are involved in the formation of LUTO, respectively PUV. This observation is in line with the previous reports of an enrichment of genomic duplications in LUTO patients.
We propose that systematic molecular karyotyping in larger LUTO cohorts will identify further causative CNVs of which a substantial proportion will be in origin de novo. Due to this in most cases male-limited phenotype, maternal transmission from healthy mothers must be kept in mind.”
Comment 5:
It is not clear why a historic reference from 1969 is used as a reference 1 for a claim that posterior urethral valves are the most common form of LUTO. Abeit not a mistake per se, a newer reference would be more fitting in my opinion.
Answer to Comment 5:
We agree with the reviewer and added a reference referring to a more recent work as reference number 2.